# Learning by Co-Designing a Board Game to Make Chain Store Knowledge More Robust

**Kuan-Yin Lee [1], Yu-Hsin Chang [1] and Prasana Kumar Samanta [2],***

[1] Department of Marketing and Logistics Management, Chaoyang University of Technology, Taichung City 413310, Taiwan
[2] Department of Business Administration, Chaoyang University of Technology, Taichung City 413310, Taiwan
* Correspondence: prasanna.samanta.777@gmail.com

**Abstract:** The modern student is used to visual information and needs an engaging, stimulating, and fun method of teaching to make learning enjoyable and memorable. Recently, more and more teachers are changing traditional teaching methods and incorporating the concept of learner-centered teaching into their courses. Students must actively identify gaps in self-knowledge, construct clear learning topics, and then integrate relevant information to explain or solve problems. In order to enhance students' interest in learning and affect their learning effectiveness, the present study introduces students to problem-oriented and game-based learning methods for solving the development problems of chain board games. Students in the third year already possess basic theoretical knowledge and have achieved relevant learning achievements, such as competition awards, industry/academic experience, class cadre experience, community service, etc. Thus, 125 students from two classes participated in this study via quantitative questionnaires. Data analysis with SPSS data revealed significant differences between learning effectiveness and learning methods, social interaction, and subject engagement; students with good learning outcomes were significantly more likely to organize notes and use methods than those with poor learning outcomes.

**Keywords:** game learning method; curriculum design; learning engagements; board game; learning effectiveness

## 1. Research Motivation and Purpose

To improve learning outcomes, effective teaching strategies should be developed to address changes in the educational scene [1]. According to Razon [2], play is an activity that can create happiness when it is freely performed. As a result of playing games, children can increase their abilities, improve their mood, and stimulate their growth. Through game-based learning with board games, students experience more fun and gain more enthusiasm for learning. According to a study by [3], educational games offer students repeated opportunities for decision-making and action. Students can learn teamwork through educational games, which develop their decision-making, design, strategy, cooperation, and problem-solving abilities [4]. By developing games, educators can convey knowledge through game development, and learners will be able to connect with the course content to demonstrate knowledge conversion through the behavioral display. As a result, games can allow the present generation to find meaning and fun within a complex interconnected knowledge base. At the same time, more and more teachers are utilizing learner-centered teaching concepts to change traditional teaching methods in their courses. To promote active learning, they design teaching activities that motivate students and arouse their interest in learning. When using problem-based learning (PBL), students identify gaps in their knowledge, create specific learning topics, and integrate relevant information into explanations or solutions. When "games" are an integral part of modern life, they can provide entertainment, unite people, and sometimes even serve as a method of education. Board games have become increasingly popular in recent years. Board games allow people

around the world to connect both through the "Internet" and enjoy games in "face-to-face" contact, and even enjoy the emotional interaction between people and with electronic products. There is a striking contrast between age groups. When reviewing our classroom, in order to enhance students learning efficiency and understand how to get the students' attention, the following issues were found:

(1)     In the digital era, the mobile phone is always with students;
(2)     A shorter concentration time is observed every year;
(3)     There are still small groups in the class that are difficult to reach even in an era of cross-domain learning, multiculturalism, and swaying youth;
(4)     There seems to be a lack of fulfillment of ambitions and talent among the learners, and the teacher does not seem to comprehend what they are trying to achieve.

Therefore, we aimed to understand what could be done to bring about the following issues:

(1)     The interaction will be brought back to the human world through board games, while at the same time, teachers will have the opportunity to teach with the help of board games;
(2)     Human-to-human interaction can be enhanced by sharing and learning from the results;
(3)     Develop chain knowledge into related board games and develop learning capabilities among students through the board game development program.

The purpose of this project was to design a unique and complete board game guided by students who study chain enterprise management and to let their peers test out the game. Students can learn and develop interpersonal relationships through board game activities, which can activate their brains, increase their interest in life and use their brains for learning.

Using problem-based learning (PBL), teachers teach students about chains and guide them in designing a chain enterprise board game, and then build soft skills by learning content about chain enterprise management. Furthermore, lecturers from the board game industry are invited to teach students how to develop board games based on theoretical foundations.

The student should regain control of the learning process, actively gain knowledge, and set a goal to develop the chain board games. The learning process requires students to integrate knowledge, and consider factors such as entertainment, ease of play, and comprehension, so their peers can also have fun learning.

The specific research questions (RQs) of the study are:

RQ1: To test whether there is a difference between students' learning effectiveness, learning engagement, emotional engagement, and interactive engagement through game-based participatory learning.

RQ2: Examine whether there is a difference between the ratings of board games designed through game-based participatory learning, the ratings after inviting peers to play, and learning outcomes, learning engagement, emotional engagement, and interactive engagement.

*Research Purposes*

A study of PBL suggests that, through this teaching method, students can become better at solving problems and managing related decisions, cultivating active learning attitudes and skills, strengthening knowledge application and memory, and cultivating teamwork [5].

The goal of the present study is to help students learn by doing by using "problem-oriented and game-based learning methods to solve chain board game development problems", thereby increasing their interest in learning and improving the effectiveness of their learning.

## 2. Literature Review

### 2.1. Development of Board Game

Gameplay enhances interpersonal relationships, advances education and learning, and improves learning stability. Games cultivate creativity, and emotional management, and enhance interpersonal relationships [6]. Many theories can be used to argue for games being an effective learning tool. According to the ARCS motivation theory, games enhance learning motivation by increasing attention, relevance, confidence, and satisfaction [7]. Contextual learning theory emphasizes the importance of educational environments and contexts for influencing authentic learning activities and knowledge [8]. Therefore, students will be motivated to learn through games, allowing them to play within the context of games, which will revolutionize current education. In this way, learners will be able to focus and become more engaged.

Several studies have proved that board games can be a very useful tool for improving education efficiency. An educational board game with mobile and sensor technologies can assist students in learning [9]. A board game was used by [10] to improve students' English dialogue skills. Chiang, Wang, and Tang [11] developed a educational board game about nutrition to improve students' knowledge of nutrition. Teachers can use board games to increase their students' engagement in classrooms by using them as educational tools [12]. Furthermore, Kafai [13] proposes that teachers can guide the learners to create their games for learning. Azizan [14] instructed the students to develop a board game and embedded technical-based questions as a cooperative learning strategy. Therefore, this research work aims to guide students to design a board game and learn relevant knowledge through the process.

### 2.2. Learning Engagement

Students need to be involved in learning, according to Reeve and Tseng [15], and they should not just be passive recipients of the information. Increasing student engagement will lead to better learning outcomes and will also positively impact teachers' teaching practices [16].

Norris, Pignal, and Lipps [17] define behavioral engagement as taking part in related activities inside and outside the classroom, whereas emotional engagement refers to school identification, belonging to the school, liking the school, or being bored at the school [18]. Tadesse & Edo's [19] study identifies statistically significant relationships between factors affecting student engagement and learning outcomes. Game development is the primary focus of this research; not the development of commercial games but the development of games as a learning method. The real purpose is to apply the knowledge learned in game development. It is therefore important to consider the learning engagement when developing games. By interacting well with teachers and peers, the emotional side can help the game learning curriculum achieve its purpose. Hence, this study examines student learning engagement through study skills, emotional engagement, and interactive engagement.

### 2.3. Learning Outcomes

To determine learning effectiveness, students' academic performance (midterms and final grades) must be measured [20]. There are two types of learning outcomes: cognitive learning and perceptual learning. According to Siegler [21], cognitive learning involves changes in personal psychology, while perceptual learning involves changes in learners' perceptions of their skills and knowledge. In this study, the learning effect is defined as the subjective effect a participant experiences after learning from the competition.

Students will be more engaged in course participation when effective teaching methods are used [22]. To achieve high learning engagement, high levels of learning, emotional engagement, and interactive engagement need to be provided to students. In the past, courses were judged by exam scores. Is it possible to determine the learning effect of students by observing the gains in learning rather than only by looking at the exam results?

This study examines whether a group's participation in learning through games and peer evaluations after trial play leads to a positive relationship between students' self-evaluated learning outcomes, engagement in learning, emotional engagement, and interaction with the game. Is there a difference between emotional engagement and interactive engagement?

## 3. Curriculum Design and Research Methods

### 3.1. Curriculum Design

Course Design Content

To verify whether the development of a board game can improve the learning input and learning effect of college students, this research selected a 3-credit chain enterprise management course, with a total course time of 3 h between weeks 2 and 8, including 1.5 h for introducing the course content, and 1.5 h to experience board games and learn the course content through participatory learning. The course design is as follows, mentioned in Table 1:

**Table 1.** Chain enterprise management course design.

| Weekly | Content | Teaching Activity |
|--------|---------|-------------------|
| Week 01 | Course Introduction | 1. The teacher explains the course objectives, teaching methods and assessment methods<br>2. Introduction to PBL<br>3. Group into groups of 5–6 people |
| Week 02 | 1. History and introduction of chain enterprise development | |
| Week 03 | 2. Chain enterprise market opportunities | |
| Week 04 | 3. Chain headquarters management, organization and development conditions | 1. Teachers will teach the theoretical knowledge related to chain enterprises |
| Week 05 | 4. Operation and management of chain headquarters | 2. Read the case and discuss it in a group<br>3. Share the outcome of the discussion |
| Week 06 | 5. Evaluation of joining business opportunities | 4. Board game experience |
| Week 07 | 6. Franchise store operation strategy | |
| Week 08 | 7. Relationship between franchisees of chain headquarters | |
| Week 09 | Board Game Development and Design Speech | 1. Board game development process sharing<br>2. Board Game Setting Mechanism |
| Week 10 | Work Discussion | |
| Week 11–12 | Work published | Show board game results<br>Lead other group members to play self-designed games<br>Board game development |
| Week 13 | Learning reflection | |

### 3.2. Game Experience Process

In this study, students learned, designed, and experienced board games through seven steps. The first step is the course description. The first week of this course explains how the course is conducted. At the end of the chain management course, students have to create a tabletop game based on the knowledge they gained. The purpose of this course was to make learning more effective through games, not to commercialize the created games. Therefore, this course is focused on integrating theoretical knowledge into the game. This study conducted 18 weeks of course teaching and completed the grouping of courses in 1–2 weeks of class.

The second step is to choose the chapter's focus. This study plans the key points of the course chapters taught every week and invites teachers with more than ten years of experience in practice and teaching chain courses to review and check back and forth to ensure the validity of its content. The content of each chapter is shown in Appendix A.

The third step is to provide a board game experience. Since the aim of this research is to teach by playing and designing board games, we selected low-difficulty board games to make the learning process easier for students. We provided a total of 11 sets of board games for 5–12-year-olds for group students to try.

The fourth step is to select the tabletop game content. The teacher lists the key points of each chapter for the group to check. Each chapter must select at least one key point of knowledge about chain businesses, and each chapter can only be selected by three groups at most, which will then be used as the core of the development of board game content.

The fifth step is board game discussion and revision. Teachers and students confirm the application of knowledge by designing a set of board games. At the same time, students make the instruction manual of the board game and shoot the instruction video of the board game.

The sixth step is to try it out with the other students. The teacher arranges a board game trial between groups. The principle of arrangement is: (1) Arrange for each group to try board games with different chapters from their group; (2) Each group plays at least 3 games, practicing chain management knowledge with different chapters, to achieve the course's purpose of game learning.

The seventh step is peer evaluation. During this study, students fill out a questionnaire immediately after trying out the board games of each group. It will ensure that the answers are accurate and not biased by other groups. The questionnaire included three parts: (1) Scoring the chain issues of this group, described as "Through playing the board game, I acquired knowledge of the chain of '###'"; (2) The ease of play of the board game, (3) Once all the group board games have been completed, complete the questionnaire about learning skills, emotional input, interactive input, and effectiveness of learning. Figure 1 shows the flow chart of board game experience and development.

In the present study, board games are considered a useful and efficient way to enhance students' interest in learning. According to Booth [23], we are in the midst of a board game renaissance. In this study, board games are mainly associated with the context of involving students in the courses which proves to be very useful in improving students' overall knowledge about the course.

During a board game, players move pieces according to specific patterns marked on the board [24]. The playing of board games may help children to learn the importance of following rules and staying seated for a certain period. Children's concentration levels may also be increased [25]. The use of board games can boost students' motivation and interest in education by stimulating their interests.

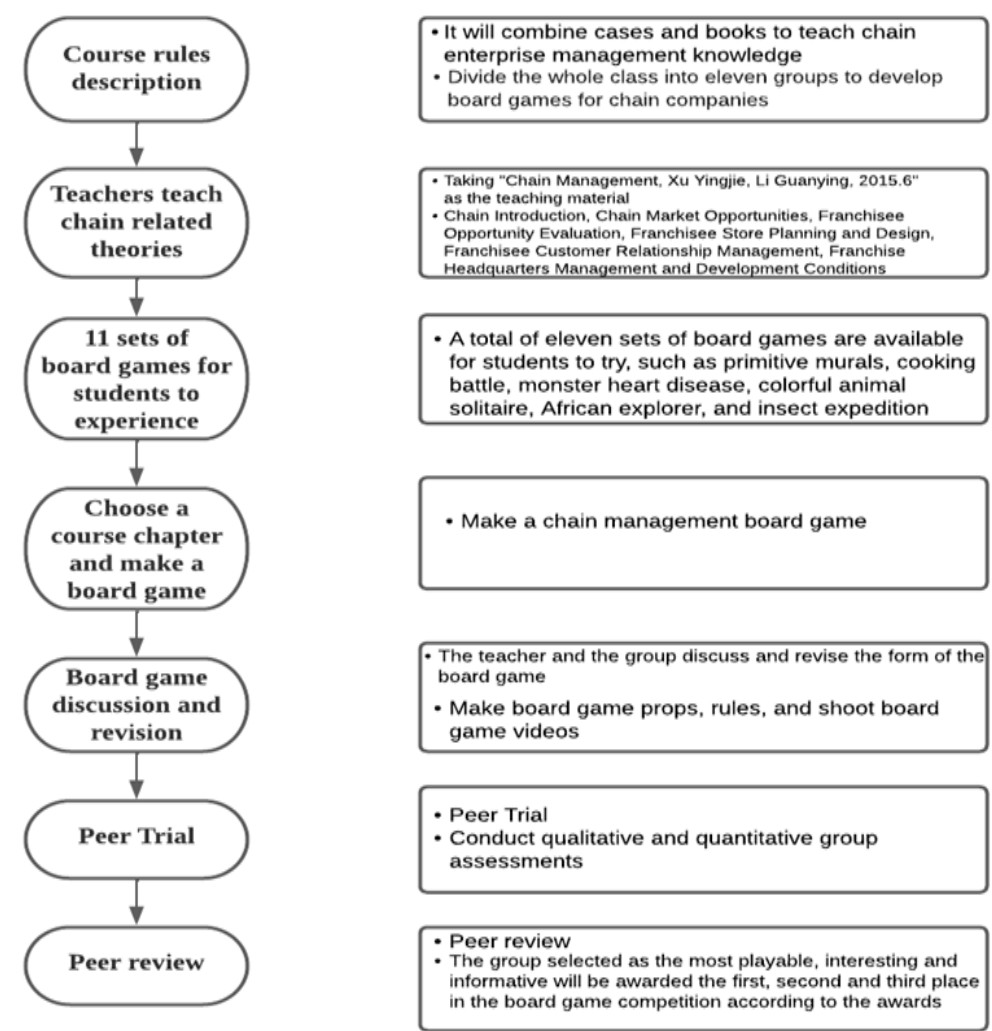

**Figure 1.** Board game experience and development flow chart.

### 4. Research Methods

*4.1. Research Object*

The participants in this research were junior students. Since they already have basic theoretical knowledge and have accumulated relevant learning achievements, they have a certain degree of experience in the application and discussion of learning. A total of 125 students from two classes were invited to participate in the study.

*4.2. Research Tools*

Several scales from previous education studies were used to measure all constructs (see Table 2). According to Table 2, the reliability of the four constructs was considered acceptable as all reliability coefficients were above the recommended value of 0.7. The factor loadings for all constructs were above 0.5. The results suggest that these constructs are reliable and unidimensional [26,27].

From the results of the table as mentioned in Table 3, it can be seen that the level of learning effectiveness is related to the student's learning engagement (F value = 7.43, $p < 0.05$), emotional engagement (F value = 8.65, $p < 0.05$), and interaction engagement (F value = 3.37, $p < 0.1$), with significant differences. Students with high learning outcomes have better performance at learning skills, emotional engagement, and engagement.

The research involved two chain management classes. Each class was divided into 11 groups, with a total of 22 groups. The course produced 22 sets of board games, one for each group of students.

**Table 2.** Source and reliability of facet items, factor analysis.

| Facet | Question Item | Factor Loading | Cronbach Alpha | Reference Source |
|---|---|---|---|---|
| Learning engagement | I will organize my notes carefully, to remember the key points of the chapter | 0.66 | 0.81 | [28] |
| | I will use the methods and knowledge I have learned to complete the homework | 0.81 | | |
| | I can highlight the key points of the textbook content | 0.83 | | |
| | I will use various methods to understand the content of the teacher's lectures | 0.71 | | |
| Emotional engagement | I get along well with my team members at school | 0.62 | 0.73 | |
| | I get along well with the teacher | 0.79 | | |
| | The teacher respects me very much | 0.82 | | |
| Interactive engagement | During class, I will take the initiative to ask questions | 0.85 | 0.74 | |
| | When I was discussing in class, will actively express opinions | 0.87 | | |
| learning outcomes | Board game design improves my learning motivation | 0.85 | 0.70 | [29] |
| | Board game design improves my learning efficiency | 0.82 | | |
| | Board game design enables me to understand chain knowledge faster | 0.65 | | |

**Table 3.** The average of learning engagement, emotional engagement and interactive engagement on learning outcomes, ANOVA.

| Facet | Learning Outcomes | Average | Variation Homogeneity Test | ROBUST Test | F Value |
|---|---|---|---|---|---|
| Learning engagement | High | 4.19 | | 7.40 (0.00) ** | 7.43 (0.00) ** |
| | Low | 3.94 | | | |
| Emotional engagement | High | 4.69 | 2.30 (0.13) | | 8.65 (0.00) ** |
| | Low | 4.47 | | | |
| Interactive engagement | High | 3.3 | | 3.35 (0.070) * | 3.37 (0.069) * |
| | Low | 3.03 | | | |

\* $p < 0.1$; \*\* $p < 0.05$.

When each group completes the design of the chain knowledge board game, other groups are invited to try it out. After the trial play, the students will rate the group to understand the extent to which the students have acquired chain knowledge through the board game. Table 4 shows the chain knowledge corresponding to the board game, and the average score after trial play with other groups. The average score ranged from 2.86 to 4.6, and the average score was 4 points.

We further conducted an ANOVA analysis of the average level of board games and learning skills, emotional engagement, interactive engagement, and learning outcomes as mentioned in Table 5. The results of the analysis found that the average score of board games had a significant effect on students' interaction engagement (F value = 3.71, $p < 0.1$). There was no significant difference in effectiveness (F value = 7.43). Therefore, from the results of this study, it can be found that the group with a low board game score was better than the group with a high board game score in terms of interaction engagement. Despite

higher peer ratings, their level of interaction was lower than that of their lower-rated peers. However, no matter how high or low the board game score is, it does not affect students' learning outcomes.

**Table 4.** The chain knowledge corresponding to the board game, and the average score of the trial.

| | Class A | | | | Class B | | |
|---|---|---|---|---|---|---|---|
| Board Game Name | Chapter | Chapter Focus | Average Score | Board Game Name | Chapter | Chapter Focus | Average Score |
| I am the big boss | 1, 6 | 1-3, 6-2 | 4.04~4.09 | Secret code | 5, 6 | 5-2, 6-2, 6-3 | 3.69 |
| Entrepreneurial journey | 3, 6 | 3-2, 3-4, 5-2 | 4.04–4.16 | Chain godfather | 2, 3, 6 | 2-1, 2-2, 2-3, 3-3, 6-1, 6-3 | 3.79–4.21 |
| Chain explorer | 1 | 1-2, 1-3, 1-4 | 2.86~3.19 | OH-YEAH | 4, 5 | 4-1, 4-2, 5-1, 5-2 | 3.1–3.8 |
| Franchise franchises and shop windows in pairs | 2, 6 | 2-1, 2-2, 6-1 | 3.14–3.64 | Bang bar chain | 1, 3 | 1-1, 1-2, 1-3, 1-4, 3-1, 3-2, 3-3, 3-4 | 3.93~4.53 |
| Franchise battle | 1, 3 | 1-1, 1-3, 3-1, 3-3 | 4.17–4.52 | Chain Monopoly | 1, 3 | 1-1, 1-2, 3-2 | 3.71~3.76 |
| I really want to eat fruit | 6 | 6-3 | 4.13 | Bomb drop card | 1, 3 | 1-1, 1-2, 1-3, 1-4, 3-1, 3-2, 3-3, 3-4 | 4 |
| Heart disease chain | 2, 3 | 2-2, 2-3, 3-2, 3-4 | 3.65–4.22 | Wilderness battle | 1, 2 | 1-3, 1-4, 2-1, 2-2 | 3.6~4.53 |
| BOM | 2, 4 | 2-3, 4-1, 4-2 | 3.43–3.52 | Chain tycoon | 1, 2 | 1-1, 1-2, 1-3, 2-1, 2-3 | 4.4–4.6 |
| I want to be the ally | 1, 4 | 1-2, 1-3, 4-2 | 3.74–4.52 | Mind Warfare | 2, 3 | 2-1, 2-2, 2-3, 3-1, 3-2, 3-3, 3-4 | 3.93~4.2 |
| Fruit Pai Pai | 4, 6 | 4-1, 4-2, 6-1 | 4–4.3 | Who is a franchisee | 4, 5 | 4-3, 5-2 | 3.2~3.38 |
| Land bidding | 3, 5 | 3-4, 5-1 | 3.87–4.09 | Franchise owner's struggle | 4, 6 | 4-1, 4-2, 6-3 | 3.7~4.2 |

**Table 5.** Means of Learning engagement, Emotional Engagement, and Interactive Engagement in Board Game Ratings, ANOVA Test.

| Facet | Board Game Rating | Average | Variation Homogeneity Test | ROBUST Test | F Value |
|---|---|---|---|---|---|
| Learning engagement | High | 4.09 | | 3.68 (0.06) | 0.38 (0.54) |
| | Low | 4.03 | | | |
| Emotional engagement | High | 4.63 | 2.03 (0.16) | | 2.81 (0.1) |
| | Low | 4.5 | | | |
| Interactive engagement | High | 3.19 | 0.89 (0.35) | | 3.55 (0.06) * |
| | Low | 3.44 | | | |
| Learning outcomes | High | 4.3 | 0.04 (0.84) | | 0.42 (0.20) |
| | Low | 4.18 | | | |

\* $p < 0.1$.

## 5. Conclusions

The present study aims to improve student learning effectiveness by designing and playing board games. It also aims to understand whether there are differences in students' learning engagement, emotional engagement, and interactive engagement.

When students are accustomed to receiving all kinds of information through their mobile phones, traditional teaching such as "chalk and talk" can no longer arouse students' interest. Instead, it needs an engaging, stimulating, and fun teaching method to drive learn-

ing. As a result of the structure of university education, teachers can customize curriculums depending on the characteristics and background of students and incorporate game-based learning connotations. Therefore, in response to changes in the educational scene, the design of effective teaching strategies to improve learning outcomes has been verified in this study [1]. This study stimulates students' enthusiasm for learning by enhancing students' experience and fun through the game-based learning of board games. However, unlike previous studies, this study allowed students to acquire franchise knowledge through the process of board game design. In addition, students can re-learn course material through the process of playing games in groups. Students learn better through this game process.

This study confirms that there are significant differences between learning effectiveness and learning engagement, emotional engagement, and interactive engagement. Students with good learning outcomes have better learning engagement, emotional engagement, and interactive engagement than students with low learning effectiveness. Students were encouraged to use the game as an opportunity to test their current understanding of how to work safely with their peers. This is supported by Rowntree [30], who lists six criteria for the evaluation of teaching media, one of which is that students should be able to recall earlier learning. Therefore, this result is consistent with previous research results. Consequently, teachers can create a good learning atmosphere in the classroom, and mutual respect and listening between students can also improve learning performance. Finally, teachers can add points to encourage students to ask questions, which is conducive to the effectiveness of the learning process.

In addition, this study is different from previous studies in that it does not require students to design a set of rigorous games; nor does it require students to play a set of well-designed board games to learn about chains. This study mainly involves students participating in the design of the game and playing and scoring other games. During the process of the game, they continuously review knowledge and achieve the best learning effect by participating in autonomous interaction. This learning effect has also been verified in this study. Especially in this study, it is found that although the grades of chain board games are different, there is no difference in the learning effect of students regardless of whether the scores of board games are high or low. This result also shows that, although the board game design did not get affirmation or high scores from other groups due to various factors, the process of designing and sharing by students is good for the overall learning effect. Students will learn more effectively if they are under less stress in the classroom. Through board games, students will be able to learn, participate, and learn about the course. Students will be more engaged in a course with this approach because it is efficient and interesting.

Grading does not determine the actual knowledge of a student in class, so it is irrelevant what grades a student receives. In this course, students gain knowledge and enhance their abilities, as their learning outcomes are impressive. At the same time, in the culture of Eastern society, students are shy to express their thoughts in public, but with the development of social media, they seek informal answers in online communities. For the teachers of the courses, it has always been difficult to grasp the learning situation of the students. In addition to tests, the teachers need to have multiple ways to understand the learning effect on the students. The results of this study found that through game-based learning, students with a low engagement in the course also performed significantly better regarding learning outcomes. Participation and investment through games is also an important method for effectively improving the learning effect.

Finally, the development of information technology has indeed not only enriched the learning resources for students but also made communication more convenient without being affected by time and space. Students are divided into small groups to play board games. After being divided into groups, they were assigned courses and played board games to learn more about them. The board games were primarily designed based on the course taught by the teacher. However, the development of information technology has also led to alienation among people. There are mainly small groups in the class, and

there is even a sense of distance between groups. Therefore, through the design of table games in this study, the groups can interact and communicate on common issues. It also allows students in each group to find warmth between people, making learning more fun than usual.

## 6. Research Limitations and Future Research Directions

The limitation of this study is that it is only applicable to chain management courses, even though it confirms that game learning has different levels of learning effectiveness, emotional engagement, and interactive engagement. This research method may apply to other business management courses, enabling students to achieve better learning outcomes through the use of participatory games. Nonetheless, more evidence is necessary to apply this method to all business management courses. Future research can examine the effects of different games on learning outcomes by integrating different games in game-based learning situations.

In addition, this study focuses on understanding the differences in re-learning input between students with high and low learning outcomes in the class and understanding students' learning outcomes through designing games and sharing games. The experimental design method can be used as a reference for teachers preparing courses in the future to investigate whether game learning and non-game learning have different effects on learning.

**Author Contributions:** Conceptualization, K.-Y.L.; Methodology, K.-Y.L.; Validation, K.-Y.L.; Investigation, P.K.S.; Writing—original draft, P.K.S.; Writing—review and editing, Y.-H.C. All authors have read and agreed to the published version of the manuscript.

**Funding:** This research received no external funding.

**Institutional Review Board Statement:** The study did not require ethical approval.

**Informed Consent Statement:** All participants in the study were students and informed consent was given prior to the questionnaires.

**Data Availability Statement:** Not applicable.

**Conflicts of Interest:** The authors declare that the research was conducted in the absence of any commercial or financial relationships that could be construed as a potential conflict of interest.

## Appendix A

**Table A1.** Key points of each chapter of chain management.

| Chain Chapters | Focus | Chain Chapters | Focus |
|---|---|---|---|
| 1. Development History and Introduction of Chain Enterprises | 1-1 Chain definition<br>1-2 Types of chain operations<br>1-3 Chain fee (three gold)<br>1-4 Chain enterprise value chain | 5. Affiliate Entrepreneurial Opportunity Assessment | 5-1 Startup Funding Rule of Thirds<br>5-2 Affiliate Survey |
| 2. Chain Enterprise Market Opportunities | 2-1 Advantages and disadvantages of chain headquarters<br>2-2 Advantages and disadvantages of franchisee<br>2-3 Chain development trend | 6. Franchise store operation strategy | 6-1 Window classification<br>6-2 Magnet theory<br>6-3 Display method<br>6-4 VP PP IP |
| 3. Chain headquarters management, organization and development conditions | 3-1 Chain Organization Chart<br>3-2 Affiliate type<br>3-3 Chain headquarters have the conditions<br>3-4 Member's own conditions | 7. The relationship between franchisees of the chain headquarters | 7-1 Way of communication<br>7-2 Chain headquarters training content<br>7-3 The basic content of the franchise company |

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
