# Peer review of "Learning by Co-Designing a Board Game to Make Chain Store Knowledge More Robust"

_education, doi:10.3390/educsci13040391_

Round 1
Reviewer 1 Report
I think this is an interesting and relevant approach to teaching with Games as a tool for review and reinforcement of course content. The paper needs a good bit of editing for clarity. There are parts that feel repetitive and other areas where the information needs additional clarification. I have tried to highlight and add notes on the PDF and hope this is helpful to you as you rework some sections of the paper. Best of luck, I hope to read your final version.

Author Response
Dear Education Science Editorial Office
Manuscript Number: education-2240767
Title: Learning by Co-designing a Board Game to make chain store knowledge more robust
Journal: Education Science
Thank you very much for offering us the chances for revision and chances for publishing in the valued journal: Education Science
We would like to express our great appreciation to editors/reviewers’ valued comments, so that we have the chance to update the manuscript to the publishable level in your precious journal.
Kindly let us know if any other questions. We are willing to follow up on any instructions for revising the manuscript according to the reviewers and editor boards’ opinions. And following is the content of the revision in accordance with the reviewers’ comments point by point.
Please kindly note for following responses:
Texts highlighted in black color represents of reviewers’ comment content.
Texts highlighted in blue color represents of our responding content
Best Regards.
Authors
We thank the editor and the reviewers for giving the author opportunity to improve the manuscript with major revisions. We appreciate their thoughtful critical comments and suggestions given to the manuscript. We realize that it is in the purpose to the improvement of the manuscript. Based on their comments, we did our best to improve the manuscript accordingly. Motivated by the reviewers’ advice, we have revised the manuscript and highlighted the changes within the manuscript. A point-by-point reply to the reviewers’ comments is attached below. We hope that you will find our responses and the revised manuscript satisfactory.
Responses from the authors
Thank you for the reviewer’s valued comments, we have updated the manuscript according to the reviewer’s comments as follows.
|
Comments from Reviewer 1 1. Since the third-year students already have basic theoretical knowledge and have accumulated relevant learning achievements, such as competition awards, industry-academia experience, class cadre experience, community experience, etc.- Sentence doesn’t seem complete
Response from the authorsStudents in the third year already possess basic theoretical knowledge and have achieved relevant learning achievements, such as competition awards, industry/academic experience, class cadre experience, community service, etc. thus, 125 students from two classes participated in the study via quantitative questionnaires. |
|
Comments from Reviewer 1 2. While at the same time, more and more teachers are employing learner-centered teaching concepts in their courses to change traditional methods of teaching- Wording problems Response from the authorsWhile at the same time, more and more teachers are utilizing learner-centered teaching concepts to change traditional teaching methods in their courses. |
|
Comments from Reviewer 1 3. Consider defining some of these terms like 3c products like 3c products
Response from the authors3c products has been changed to electronic products. |
|
Comments from Reviewer 1 4. Is these findings from this study? If not, cite.
Response from the authorsAge becomes a striking contrast. In reviewing the teaching site, in order to enhance students learning efficiency, how to get the students’ attention, the following study issues that happened in our classroom were found. (1) In the digital era, the mobile phone is always with students. (2) A shorter concentration time is observed every year. (3) There are still small groups in the class that are difficult to cross even in an era of cross-domain learning, multiculturalism, and swaying youth; (4) There seems to be a lack of fulfillment of ambitions and talents among the learners, and the teacher does not seem to comprehend what they are trying to achieve.
|
|
Comments from Reviewer 1 5. Did it not say earlier that board games connect through the internet?
Response from the authors Therefore, if we can do something in our classroom, will following study issues will happen? (1) It means that at this age student always play games in the mobile phone so as a teacher we always expect the students to keep their concentration in the class therefore the board game has been designed so that the students don’t feel boring in the class and can learn many things about the course.
|
|
Comments from Reviewer 1 6. Formatting changes?
Response from the authors
It comes under research purpose and motivation so it is being given as subtopic of 1 that is 1.1 and it is also changed to italic style.
|
|
Comments from Reviewer 1 7. This feels like a big jump. Maybe just tone down the wording? .
Response from the authors Several studies proved that board game can be a very useful tool for improving education efficiency. An educational board game with mobile and sensor technologies can assist students in learning [9]. A board game was used by [10] to improve students' English dialogue skills. Chiang, Wang, and Tang [11] developed a nutrition educational board game to improve students' nutrition knowledge. Teachers can use board games to increase their students' engagement in classrooms by using them as educational tools [12]. Furthermore, Kafai [13] propose that teacher can guide the learners to create their own games for learning. Azizan [14] instructed the students to develop a board game and embedded technical based questions as a cooperative learning strategy. Therefore, the aim of this research work is to guide students to design a board game, and learn relevant knowledge through the process..
|
|
Comments from Reviewer 1 8. Well stated! Very clear. What do you mean by this?
Response from the authors Norris, Pignal, and Lipps [17] define behavioral engagement as taking part in related activities inside and outside the classroom, whereas emotional engagement refers to school identification, belonging to the school, liking the school, or being bored at the school [18]. Tadesse & Edo's [19] study identifies statistically significant relationships between factors affecting student engagement and learning outcomes. Game development is the primary focus of this research, not the development of commercial games but the development of games as a learning method. The real purpose is to apply the knowledge learned in game development. It is, therefore important to consider the learning engagement.
|
|
Comments from Reviewer 1 9. What is the output here?
Response from the authors Here exam’s results are mentioned as the output |
|
Comments from Reviewer 1 10. Is this a RO? Seems misplaced. These are normally Called RQ research questions and should be phrased as questions
Response from the authors These are changed to research questions in the manuscript. |
|
Comments from Reviewer 1 11. avoid using conjunctions
Response from the authors At the end of the chain management course, students have to create a table game based on the knowledge they gained. Here the grouping of courses means the students are divided into different groups and they have been assigned some courses and the students tries to complete the courses by playing board games. |
|
Comments from Reviewer 1 12. Did they need to be taught to play games? Or taught to build them?
Response from the authors The third step is to provide a board game experience. Since the aim of this research is to teaching by playing and designing board games, we selected low-difficulty board games to make the learning process easier for students. Therefore, this study provides a total of 11 sets of board games for 5-12-year-olds for group students to try. . |
|
Comments from Reviewer 1 13. This was also stated earlier in the paper. I am not sure what these refer to or how they are relevant learning experiences
Response from the authors This sentence is removed from this particular part of the manuscript.
|
|
Comments from Reviewer 1 14. Which one? Be specific.
Response from the authors The table 4 shows the chain knowledge corresponding to the board game, and the average score after trial play with other groups. The average score ranges from 2.86 to 4.6, and the average score is 4 points. |
|
Comments from Reviewer 1 There are a lot of terms around this in literature. Be careful conflating game-based learning, PBL, gamified, etc. They each have unique meanings.
Response from the authors The present study aims to improve student learning effectiveness through designing and playing board game. It also aims to understand whether there are differences in students' learning engagement, emotional engagement, and interactive engagement.
|
|
Comments from Reviewer 1 This is a large statement about this study or is it a referenced study? There have been other studies that used student-designed board games
Response from the authors When students are accustomed to receiving all kinds of information through their mobile phones, traditional teaching such as chalk and talk can no longer arouse students' interest. Instead, it needs an engaging, stimulating, and fun teaching method to drive learning. As a result of the structure of university education, teachers can customize curriculums depending on the characteristics and background of students and incorporate game-based learning connotations. Therefore, in response to changes in the educational scene, the design of effective teaching strategies to improve learning outcomes has been verified in this study [28]. This study stimulates students' enthusiasm for learning by enhancing students' experience and fun through the game-based learning of board games. However, unlike previous studies, this study allowed students to acquire franchise knowledge through the process of board game design. In addition, students can re-learn course material through the process of playing games in groups. Students learn better through this game process. This study confirms that there are significant differences between learning effectiveness and learning engagement, emotional engagement, and interactive engagement. Students with good learning outcomes have better learning engagement, emotional engagement, and interactive engagement than students with low learning effectiveness. Therefore, this result is consistent with the previous research results. The same is true in the performance of an emotional investment.
|
|
Comments from Reviewer 1 Not sure what this means
Response from the authors Students will learn more effectively if they are not under more stress in the classroom. Through board games, students will be able to learn, participate, and learn about the course. Students will be more engaged in a course with this approach because it is efficient and interesting.
|
|
Comments from Reviewer 1 Was this shown in the results?
Response from the authors Grading does not determine the actual knowledge of a student in class, so it is irrelevant what grades a student receives. In this course, students gain knowledge and enhance their abilities, as their learning outcomes are impressive. |
|
Comments from Reviewer 1 How so? Were the games not played in-person?
Response from the authors Students are divided into small groups to play board games. After being divided into groups, they were assigned courses and played board games to learn more about them. It is because Board games are primarily designed based on the course taught by the teacher. |
The below references have been added in the manuscript.
- Kafai YB. Playing and making games for learning: Instructionist and constructionist perspectives for game studies. Games and culture. 2006 Jan;1(1):36-40.
- Azizan MT, Mellon N, Ramli RM, Yusup S. Improving teamwork skills and enhancing deep learning via development of board game using cooperative learning method in Reaction Engineering course. Education for Chemical Engineers. 2018 Jan 1;22:1-3.
- Booth P. Game play: paratextuality in contemporary board games. Bloomsbury Publishing USA; 2015 Apr 23.
- Kyppö J. Board games: throughout the history and multidimensional spaces. 2019 Jul 8.
- Riggs AE, Young AG. Developmental changes in children’s normative reasoning across learning contexts and collaborative roles. Developmental psychology. 2016 Aug;52(8):1236.
- Rowntree, Derek. "Educational technology in curriculum development." (1982).

Reviewer 2 Report
The core conceit of this piece is good: show that engaging students in collaborative board game design improves learning outcomes.
However a number of issues get in the way of the piece's success:
1. Writing style: There are a number of serious issues that make this piece hard to read. I strongly suggest further editing and proofreading. For example, on page 2, lines 64-65 (and a few other places), the piece slips into second person. Similarly, on page 2, at line 48, "3C" is mentioned without explanation, and at line line 62, "chain knowledge" is also introduced without explanation. These kinds of inconsistencies indicate that further revisions are needed. The specific examples described here are not the only issues, I mentioned them only as an example.
2. Incorrect use of gamification -- as Groh (2012) defines it, gamification is "the use of game design elements in non-game concepts." So, for instance, if the classroom in this study gave students achievement plaques for designing their game within certain, optional constraints, pitted students against one another in competitions, created a course-wide leaderboard, etc. that would be gamification; asking students to design a game in class is not.
Groh, F. (2012). Gamification: State of the art definition and utilization. Institute of Media Informatics Ulm University, 39, 31.
3. Limited engagement with "board games" as a concept. The board games used in the classroom are mentioned only briefly (in Figure 1 on page 6), and it is unclear if they are being described in a general sense, or if their titles are being mentioned. At a bare minimum, cite board games as you would other media, and describe what specifically about these games recommended them for classroom use. To deepen engagement with board games, authors should consider contemporary board games scholarship, e.g. Booth (2015, 2021) and the literature around board game design (e.g. Davidson & Costikyan, 2011; Engelstein & Shalev, 2022).
Booth, P. (2015). Game play: paratextuality in contemporary board games. Bloomsbury Publishing USA.
Booth, P. (2021). Board Games as Media. Bloomsbury Publishing USA.
Davidson, D., & Costikyan, G. (2011). Tabletop: Analog Game Design. Lulu. com
Engelstein, G., & Shalev, I. (2022). Building Blocks of tabletop game design: An encyclopedia of mechanisms. CRC Press.
Author Response
Manuscript Number: education-2240767
Title: Learning by Co-designing a Board Game to make chain store knowledge more robust
Journal: Education Science
Thank you very much for offering us the chances for revision and chances for publishing in the valued journal: Education Science
We would like to express our great appreciation to editors/reviewers’ valued comments, so that we have the chance to update the manuscript to the publishable level in your precious journal.
Kindly let us know if any other questions. We are willing to follow up on any instructions for revising the manuscript according to the reviewers and editor boards’ opinions. And following is the content of the revision in accordance with the reviewers’ comments point by point.
Please kindly note for following responses:
Texts highlighted in black color represents of reviewers’ comment content.
Texts highlighted in blue color represents of our responding content
Best Regards.
Authors
We thank the editor and the reviewers for giving the author opportunity to improve the manuscript with major revisions. We appreciate their thoughtful critical comments and suggestions given to the manuscript. We realize that it is in the purpose to the improvement of the manuscript. Based on their comments, we did our best to improve the manuscript accordingly. Motivated by the reviewers’ advice, we have revised the manuscript and highlighted the changes within the manuscript. A point-by-point reply to the reviewers’ comments is attached below. We hope that you will find our responses and the revised manuscript satisfactory.
Responses from the authors
Thank you for the reviewer’s valued comments, we have updated the manuscript according to the reviewer’s comments as follows.
|
Comments from Reviewer 2 Writing style: There are a number of serious issues that make this piece hard to read. I strongly suggest further editing and proofreading. For example, on page 2, lines 64-65 (and a few other places), the piece slips into second person. Similarly, on page 2, at line 48, "3C" is mentioned without explanation, and at line 62, "chain knowledge" is also introduced without explanation. These kinds of inconsistencies indicate that further revisions are needed. The specific examples described here are not the only issues, I mentioned them only as an example.
Response from the authors The sentences lines 64-65 has been changed to the sentence “Therefore, if we can do something in our classroom, will following study issues will happen?” 3c products has been changed to electronic products. |
|
Comments from Reviewer 2 Incorrect use of gamification -- as Groh (2012) defines it, gamification is "the use of game design elements in non-game concepts." So, for instance, if the classroom in this study gave students achievement plaques for designing their game within certain, optional constraints, pitted students against one another in competitions, created a course-wide leaderboard, etc. that would be gamification; asking students to design a game in class is not. Groh, F. (2012). Gamification: State of the art definition and utilization. Institute of Media Informatics Ulm University, 39, 31. Response from the authors Development of Board Game Gameplay enhances interpersonal relationships, advances education and learning, and improves learning stability. Games cultivate creativity, and emotional management, and enhance interpersonal relationships [6]. Many theories can be used as arguments when games become an effective learning tool. According to the ARCS motivation theory, games enhance learning motivation by increasing attention, relevance, confidence, and satisfaction [7]. Contextual learning theory emphasizes the importance of educational environments and contexts as influencing authentic learning activities and knowledge [8]. Therefore, students will be motivated to learn through games, allowing them to play within the context of games, which will revolutionize current education. In this way, learners will be able to focus and become more engaged. Several studies proved that board games can be a very useful tool for improving education efficiency. An educational board game with mobile and sensor technologies can assist students in learning [9]. A board game was used by [10] to improve students' English dialogue skills. Chiang, Wang, and Tang [11] developed a nutrition educational board game to improve students' nutrition knowledge. Teachers can use board games to increase their students' engagement in classrooms by using them as educational tools [12]. Furthermore, Kafai [13] propose that teacher can guide the learners to create their games for learning. Azizan [14] instructed the students to develop a board game and embedded technical-based questions as a cooperative learning strategy. Therefore, this research work aims to guide students to design a board game and learn relevant knowledge through the process. |
|
Comments from Reviewer 2 Limited engagement with "board games" as a concept. The board games used in the classroom are mentioned only briefly (in Figure 1 on page 6), and it is unclear if they are being described in a general sense, or if their titles are being mentioned. At a bare minimum, cite board games as you would other media, and describe what specifically about these games recommended them for classroom use. To deepen engagement with board games, authors should consider contemporary board games scholarship, e.g. Booth (2015, 2021) and the literature around board game design (e.g. Davidson & Costikyan, 2011; Engelstein & Shalev, 2022).
Response from the authors In the present study board games are considered a useful and efficient way to enhance students’ interest in learning the courses. According to Booth [23] We are in the midst of a board game renaissance. In this study board games are mainly associated with the context of involving students in the courses which proves to be very useful in improving students’ overall knowledge about the course. During a board game, players move pieces according to specific patterns marked on the board [24]. The playing of board games may help children to learn the importance of following rules and staying seated for a certain period. Children's concentration levels may also be increased [25]. The use of board games can boost students' motivation and interest in education by stimulating their interests. |
The below references have been added in the manuscript.
- Kafai YB. Playing and making games for learning: Instructionist and constructionist perspectives for game studies. Games and culture. 2006 Jan;1(1):36-40.
- Azizan MT, Mellon N, Ramli RM, Yusup S. Improving teamwork skills and enhancing deep learning via development of board game using cooperative learning method in Reaction Engineering course. Education for Chemical Engineers. 2018 Jan 1;22:1-3.
- Booth P. Game play: paratextuality in contemporary board games. Bloomsbury Publishing USA; 2015 Apr 23.
- Kyppö J. Board games: throughout the history and multidimensional spaces. 2019 Jul 8.
- Riggs AE, Young AG. Developmental changes in children’s normative reasoning across learning contexts and collaborative roles. Developmental psychology. 2016 Aug;52(8):1236.

Round 2
Reviewer 2 Report
The piece has been improved by your changes! However, grammar is still an issue. Notably your research questions are not phrased as questions, but there are numerous other issues requiring editing/proofreading.
Author Response
Part 1 The research question part has been edited. We have provided the editing part below for your reference.
RQ1: To test whether game-based participatory learning affects students' learning effectiveness, learning engagement, emotional engagement, and interactive engagement?
RQ2: To Investigate whether learning outcomes, learning engagement, emotional engagement, and interactive engagement of board games designed with game-based participatory learning differ from those of games developed with peers?
Part 2
This part is about the English editing of our paper. So we are providing/uploading a proof reading certificate in pdf format